# Therapeutic Approaches with Intravitreal Injections in Geographic Atrophy Secondary to Age-Related Macular Degeneration: Current Drugs and Potential Molecules

**DOI:** 10.3390/ijms20071693

**Published:** 2019-04-04

**Authors:** Marcella Nebbioso, Alessandro Lambiase, Alberto Cerini, Paolo Giuseppe Limoli, Maurizio La Cava, Antonio Greco

**Affiliations:** 1Department of Sense Organs, Faculty of Medicine and Odontology, Umberto I Policlinic, Sapienza University of Rome, p. le A. Moro 5, 00185 Rome, Italy; alessandro.lambiase@uniroma1.it (A.L.); cerinialberto@hotmail.it (A.C.); maurizio.lacava@uniroma1.it (M.L.C.); antonio.greco@uniroma1.it (A.G.); 2Low Vision Research Centre of Milan, Piazza Sempione 3, 20145 Milan, Italy; paololimoli@libero.it

**Keywords:** age-related macular degeneration, anti-inflammatory agents, complement inhibitors, dry AMD, geographic atrophy, intravitreal injection, neuroprotective agents, non-exudative AMD

## Abstract

The present review focuses on recent clinical trials that analyze the efficacy of intravitreal therapeutic agents for the treatment of dry age-related macular degeneration (AMD), such as neuroprotective drugs, and complement inhibitors, also called immunomodulatory or anti-inflammatory agents. A systematic literature search was performed to identify randomized controlled trials published prior to January 2019. Patients affected by dry AMD treated with intravitreal therapeutic agents were included. Changes in the correct visual acuity and reduction in geographic atrophy progression were evaluated. Several new drugs have shown promising results, including those targeting the complement cascade and neuroprotective agents. The potential action of the two groups of drugs is to block complement cascade upregulation of immunomodulating agents, and to prevent the degeneration and apoptosis of ganglion cells for the neuroprotectors, respectively. Our analysis indicates that finding treatments for dry AMD will require continued collaboration among researchers to identify additional molecular targets and to fully interrogate the utility of pluripotent stem cells for personalized therapy.

## 1. Introduction

Age-related macular degeneration (AMD) is the leading cause of irreversible blindness in the elderly population, and it is defined as a chronic, multifactorial, and progressive central retinal disease. The prevalence of AMD is approximately 8.69% in the worldwide population and 12.3% in Europe [1]. There were 8.4 million AMD patients with moderate to severe vision impairment in 2015 with probable increases to 196 million in 2020 [1]. Macular pigmentation changes in the chorioretinal layers and the drusen are present at the early stage of AMD, while the advanced stages are characterized by dry and/or neovascular forms [1,2]. Choroidal neovascularization (CNV) in AMD, often accompanied by serum-hemorrhagic retinal detachment, eventually leads to the degeneration of photoreceptors [2,3]. Likewise, in dry AMD or non-exudative AMD, and then subsequently in geographic atrophy (GA), there are areas of progressive atrophy and thinning mainly involving the layers of the retinal pigment epithelium (RPE) and the underlying choriocapillaris; these modifications are a prelude to the degeneration of photoreceptors leading to irreversible loss of visual function [2,3,4] (Figure 1). The main risk factors for GA are ageing, family history, cigarette smoking, cardiovascular risk factors, previous cataract surgery, etc. [3]. Thus, the characteristics of GA with progressive and irreversible loss of retinal cells inevitably are responsible for around 20% of cases of legal blindness [1,2,3,4]. No effective treatment for GA is currently available, unlike neovascular AMD in which anti-angiogenic treatments are effective in improving visual acuity [4]. Inflammation, complement activation, oxidative stress, blood flow regulation, and reduced neuroprotection are the main pathways implicated in the progression of GA [3,4]. Currently, the only preventive option for dry AMD is the Age-Related Eye Disease Studies (AREDS) formulation, which reduces the risk of AMD progression [5]. The etiology of AMD is believed to be the result of a combination of oxidative stress, chronic inflammation, predisposing genetic and environmental factors. Oxidative stress, responsible for the production of retinal reactive oxygen species (ROS), can induct chronic inflammation and programmed necrosis in RPE cells. It is thus triggered by excessive presence of photosensitizing factors, intensive oxygen metabolism, and polyunsaturated fatty acids [6]. Genetic polymorphism has been associated with AMD disease and implies the involvement of inflammation factors, lipid metabolism, angiogenesis, and RPE dysfunction [7]. This article focuses on actual, or still ongoing, clinical trials evaluating the efficacy of intravitreal therapies in dry AMD, such as neuroprotective drugs, complement inhibitors, immunomodulatory or anti-inflammatory agents.

## 2. Methods

Our systematic literature search was conducted in PubMed, Embase, Cochrane Library, and Web to identify randomized controlled trials published prior to January 2019. The search keywords were: “age-related macular degeneration”, “atrophic”, “dry”, “current drugs”, “geographic atrophy”, “non-exudative”, “intravitreal injections”, and “potential molecules”. The patients dry AMD treated with intravitreal therapeutic agents were included. The changes in the correct visual acuity (BCVA) and the reduction in GA progression were evaluated. We also studied reviews, comments, and disquisitions on the pathology.

## 3. Main Pathogenetic Pathways in Dry AMD: Complement Dysfunction and Inflammation

The therapies target different aspects of GA, including inflammatory pathways, oxidative stress RPE degeneration, byproducts of the visual cycle, restoration of choroidal perfusion, use of growth factors (GFs), modification of cellular DNA through genetic therapy, and replenishing RPE cells with stem cell-derived RPE cells [6,7,8,9]. It has been hypothesized that anti-inflammatory agents could represent a therapeutic option [8]. In fact, chronic inflammation is reported to play an important role in early AMD pathogenesis.

The deposition of intracellular drusen in the RPE layer, containing cell debris and proteins such as complement components, triggers the inflammatory response [7,8]. The disease progresses, consequently, with sustained inflammatory response, further drusen accumulation and oxidative stress, resulting in damage and eventually cell death. Immunohistochemical examination of drusen shows many proinflammatory proteins including apolipoprotein E, acute phase proteins, coagulation proteins, β-amyloid (Aβ), and complement activation components during the development of the disease [7,8,9]. The complement cascades in its various forms, classical, alternative and mannose-binding lectin, converge on a final single pathway via the cleavage of complement factor C3 into C3a and C3b that leads to phagocytosis, inflammation, formation of a membrane attack complex (MAC) and ultimately to cell death [9] (Scheme 1). 

The alternative pathway has been implicated in the pathogenesis of the disease; in particular, complement components 2, 3, and 7 have shown to be associated with AMD. The formation of MAC consequent to the activation of the complement pathway causes cell lysis and chemokine release, thus, causing the recruitment of inflammatory cells and the increase of vascular permeability [9]. Hence, the inhibition of the overactive complement pathway represents a viable therapeutic approach to arrest the progression of GA. Moreover, there is an interest in the role of immune dysfunction, such as inappropriate complement cascade activation, in the etiology of AMD. Knowledge of the major role that complement cascade activation plays in the disease has led to several therapeutic options [9,10,11,12]. 

In summary, we know that numerous proteins and polypeptides have therapeutic effects in the treatment of degenerative diseases, but localized treatment of retinal diseases is complicated by the blood-retinal barrier that hinders the penetration of several molecules from the circulatory blood system to the neurosensory retina [3,4]. Therefore, the intravitreal injection (IVI), without systemic exposure, could be the simplest and most useful route to allow an adequate chorioretinal restoration and preserve vision [3,4]. The present review will focus on neuroprotective, and immunomodulatory, or anti-inflammatory agents that have been used for IVI in patients with GA (Figure 2). A summary of relevant clinical trials is presented in Table 1.

## 4. Potential Therapeutic Molecules in Dry AMD

### 4.1. Neuroprotective Agents Designed to Prevent Retinal Ganglion Cell Apoptosis

#### 4.1.1. Brimonidine

Brimonidine is a selective α2 adrenergic (α2A) receptor agonist that has been used for its pressure-lowering effects in the treatment of glaucoma for several years. Its mechanism in lowering the intraocular pressure involves both the decrease of aqueous humor production and increased uveoscleral outflow [13]. No significant side effects are known, although an excess of drowsiness and lethargy have been reported as side effects in small patients [35]. In animal models it was found that brimonidine protects photoreceptors, bipolar cells, and retinal ganglion cells from a variety of retinal insults, including retinal phototoxicity and retinal ischemia. The authors state that neuroprotection is achieved via different mechanisms such as the release of brain-derived neurotrophic factor (BDNF) from retinal ganglion cells and by upregulation of cell survival signaling pathways [36]. Importantly, brimonidine has been demonstrated to be neuroprotective via the modulation of Aβ toxicity. This pathway is strongly implicated in neurodegenerative conditions such as Alzheimer’s disease and in glaucoma-related models. Aβ is the major constituent of senile plaque and may, therefore, play a key role in the stress–response to induce cellular apoptosis, even though, to this day, a comprehensive mechanism is not fully understood [13].

Nizari et al. proposed a model of α2A agonists’ neuroprotective effect against Aβ toxicity [13]. Aβ is associated with abnormal processing of amyloid precursor protein (APP). APP can be metabolized into soluble APPα (sAPPα) by the non-amyloidogenic pathway or into Aβ by the amyloidogenic pathway; α2A receptor agonists have been reported to negatively affect the amyloidogenic pathway, preventing the excitotoxicity mediated by glutamate and, therefore, cell death. Moreover, α2A receptor agonists can also affect APP processing via the extracellular matrix, by modulating matrix metalloproteinase-9 (MMP-9) and laminin through *laminin*-binding protein (LBP), preventing further toxic interactions with Aβ and by increasing the processing of APP into sAPPα, promoting the non-amyloidogenic pathway. Furthermore, α2A receptor agonists can also increase levels of phosphorylated arabinose-inducible BAD promoter (P-Bad), thereby promoting cell survival and neuroprotection [13,14]. The initial results of a study (Clinicaltrials.gov: NCT00658619) involving 119 participants demonstrated that IVIs of 200 µg or 400 µg of brimonidine produced a significant reduction in GA size when compared to the sham arm. In particular, the 200 µg dose produced a reduction in GA size by 18% while the 400 µg dose produced a 27% reduction. No significant adverse effects have been reported for either injections [14]. 

A larger phase 2 trial of 310 patients is currently underway (Beacon, NCT02087085). It is a double-blind, sham-control study with its primary outcome being the changes in size of GA lesion from baseline to 24 months [14]. Participants were randomized to receive either 400 μg sustained release solid brimonidine tartrate implant administered intravitreally on day one then every 3 months through month 21 or sham via needleless applicator. This clinical trial is expected to end in 2019.

#### 4.1.2. Ciliary Neurotrophic Factor

Ciliary neurotrophic factor (CNTF) is a neurotrophic factor member of the interleukin-6 (IL-6) family of neuropoietic cytokines. It influences the survival and differentiation of cells in the nervous system, including retinal cells, although the function of CNTF is not fully understood [37]. 

Its activities are mediated through a heterotrimeric complex formed by a specific α subunit CNTF receptor (CNTFRα) and two β subunits, leukemia inhibitory factor receptor (LIFRβ) and IL-6 signal transducer (gp130). CNTF is able to delay the loss of cells during retinal degeneration by protecting the photoreceptors in 12 animal models [37]. The treatment with CNTF targets Müller glia in order to trigger a cascade of signaling events leading to photoreceptor survival. CNTF activates complex molecular and cellular responses in numerous cell types. Exposure to high levels of CNTF regulates photoreceptor specific genes, particularly those associated with phototransduction. This could profoundly influence neuronal characteristics and chorioretinal function.

Delivery of CNTF to the retina is a major challenge: encapsulated cell technology (ECT) and the NT-501 implant were developed in some studies [38]. The implant was developed to deliver the drug directly to the retina over a sustained period. The availability of the protein is assured by a stable and long-term secretion that also allows a continuous exposure of the target site. Preliminary results involving the NT-501 ECT implant appear promising, demonstrating sustained, safe and efficacious delivery of the protein therapy of up to several years in the eye. ECT is a system consisting of human cell lines that are genetically engineered to endogenously express a selected therapeutic protein at a regulated delivery rate [15,16,17,38,39]. Many studies on animal models have demonstrated the possibility of using CNTF as an approach to reduce photoreceptor cell loss. Between February and October 2007, 51 patients with GA were enrolled and randomly assigned to phase 1 and 2 of the studies to evaluate the effect on retinal structure and function (NCT00063765, NCT00447954, NCT00447980, NCT00447993) [15,18]. Patients were randomly assigned to receive a high-dose implant, a low-dose implant or sham surgery in one eye at the ratio of 2:1:1. All patients completed the 12-mo endpoint, and no patients dropped out of the study. While in the high- or low-dose CNTF groups a statistically significant difference in total macular volume was found, compared to baseline at all-time points (months 4, 6, and 12) (*p* < 0.001), instead no variation appeared in the sham one [17,37]. In particular, the high-dose group had a more remarkable outcome than the low-dose at all-time points (*p* < 0.05). As shown by cross-sectional evaluation of high-resolution line scans, these results were linked to an increased width of the external layer complex. The GA area varied slightly among the 3 groups at baseline, but the difference was not statistically significant. Although no improvement in visual acuity was observed in the three groups, an increase in retinal thickness maintained during the whole follow up period (12 mo, *p* < 0.001) was observed in the groups treated with CTNF implants. The result has been reported to be dose-dependent with better response in high-dose patients. The observed increase in retinal thickness was associated with visual acuity stabilization regardless of baseline BCVA in high-dose patients [15,18].

### 4.2. Immune Modulating or Anti-Inflammatory Agents

#### 4.2.1. Lampalizumab

Lampalizumab, previously referred to as anti-complement factor D antibody, anti-factor D, FCFD4514S, RG7417, RO 5490249, or TNX-234, is an antigen-binding fragment (Fab) of a humanized monoclonal antibody (mAb) directed against complement factor D (CFD). Lampalizumab selectively inhibits the activation mediated by the CFD of the alternative complement pathway. Lampalizumab does not act on the classical or mannose-binding lectin pathways of the complement activation [19]. The Mahalo phase 2 clinical trial (NCT01229215) investigated the efficacy of IVIs of lampalizumab in patients with GA. Moreover, the trial examined both the safety and the pharmacokinetics of lampalizumab [20]. This trial enrolled 120 patients, and it demonstrated an acceptable safety profile during the 18-month treatment period. Monthly lampalizumab treatment demonstrated a 20% reduction in lesion area progression versus sham control. A more substantial benefit from the monthly treatment (44% reduction in GA area progression compared to control) was observed in a subgroup of complement factor I (CFI) risk-allele carriers (57% of the patients analyzed were CFI risk-allele carriers). The Mahalo study, published in 2013, showed a potential effect of the treatment in patients with GA and supported therapeutic targeting of the alternative complement pathway for treating AMD pathogenesis [20]. Between August 2014, and October 2016, 906 Chroma (GX29176; NCT02247479) participants and 975 Spectri (GX29185; NCT02247531) participants randomly underwent sham injections every 4 weeks (153 Chroma and 161 Spectri, respectively), lampalizumab every 4 weeks (298 Chroma and 330 Spectri, respectively), sham every 6 weeks (152 Chroma and 160 Spectri, respectively), or lampalizumab every 6 weeks (303 Chroma and 324 Spectri, respectively) [21]. Both the Chroma and Spectri phase 3 trials showed no significant difference in GA progression during the whole follow period between sham and lampalizumpab treated arms [21]. 

#### 4.2.2. Zimura 

Zimura (ARC-1905) is a polyethylene glycol (PEG), oligonucleotide, chemically synthesized single strand nucleic acid aptamer that targets and inhibits complement factor C5. The inhibition of C5 in the complement cascade prevents the formation of key terminal fragments (C5a and C5b-9). C5b-9 is involved in the formation of the MAC, which causes cellular death through the disruption of the cell membrane [22,23,24]. A phase 1 trial for dry AMD (NCT00950638), started in 2009, was completed in 2012. It evaluated the safety and tolerability of intravitreous Zimura injections. Forty-seven participants, 50 years or older, with GA secondary to dry AMD in both eyes were recruited. The study completed with no results posted. The phase 2a trial (NCT03362190) evaluated the safety profile of Zimura administered intravitreally in combination with 0.5 mg of Ranibizumab, in 65 wet AMD patients who had not previously been administered an anti-vascular endothelial GF (VEGF) drug [25]. A considerably higher percentage of patients receiving the Ranibizumab–Zimura combination showed improved visual acuity compared with controls of patients receiving Ranibizumab monotherapy. Later, in October 2018, Ophthotech completed patient enrolment (estimated 120 patients) in its phase 2b clinical trial to evaluate the safety and efficacy of Zimura compared to sham injection in subjects with autosomal recessive Stargardt disease 1. The company has decided to modify its ongoing phase 2/3 clinical trial of Zimura monotherapy in 200 participants with GA secondary to dry AMD (NCT02686658). The trial has been adjusted to accelerate the deadline by reducing the number of patients, shortening the time for attaining the primary efficacy endpoint and thereby reducing the cost to complete the study. Estimated primary completion date is November 2019 [25].

#### 4.2.3. POT-4 and APL-2

POT-4 is a cyclic peptide comprising 13 amino acids derived from compstatin, which irreversibly binds to C3 and prevents its proteolytic activation to C3a and C3b; thus, inhibiting the complement pathways and preventing MAC formation. It forms a gel when injected at high concentrations into the vitreous. A phase I clinical trial (NCT00473928) in wet AMD patients was completed in 2010 without safety concerns at doses up to 1.05 mg [27,28,40]. The potent inhibition of this drug on the complement could increase the risk of endophthalmitis; therefore, it is necessary to establish the appropriate safety dose [40]. APL-2 is a modified version of POT-4 designed to have a longer half-life. APL-2 (POT-4/AL-78898A) is a synthetic cyclic peptide conjugated to a PEG polymer that binds specifically to C3, effectively blocking all three pathways of complement activation: classical, lectin, and alternative [26]. It has undergone phase 1 and 2 trials (NCT00473928, NCT02503332), and it showed no safety concerns. A multicenter, randomized, single-masked, sham-controlled clinical phase 2 research project, named the Filly trial, was carried on 246 patients with GA, at over 40 clinical sites, located in the United States, Australia, and New Zealand. APL-2 was injected as IVI monthly for 12 months, and the patients were followed for an extra period of 18 months [26]. Apellis Pharmaceuticals reported that APL-2, after a 12 month follow-up, determined a reduction in the GA growth rate. Every month injection reduced the GA growth rate by 29%, while every other month injection reduced GA growth rate by 20% when compared to the sham arm. After the 12-month period, subjects were followed for a further six months without treatment. During this period of non-treatment, the GA lesions in the previously treated groups grew at a rate similar to sham. Subjects previously treated with monthly APL-2 showed only a 12% reduction over the six-month period compared to sham, while those previously treated with every other month APL-2 showed a 9% reduction compared to sham. Two AMD phase 3 clinical trials, Oaks and Derby (NCT03525613-NCT03525600), are being initiated for the development of APL-2 in the treatment of GA secondary to advanced AMD in 600 participants each. A multi-center, randomized, double-masked, sham-controlled study to compare the efficacy and safety of intravitreal APL-2 therapy with sham injections the subject population will consist of subjects with GA secondary to AMD: patients’ recruitment is still ongoing, and the estimated study completion date is December 2022 [26].

#### 4.2.4. CLG561

CLG561 is an inhibitor of properdin. It acts to stabilize the alternative pathway C3 and C5 convertases by extending the half-lives of the C3 and C5 converting enzymes; based on human genetics as well as pathophysiological features of AMD that implicate complement activation [26]. A phase 1 study (NCT01835015) has ended in 2016; it has evaluated the safety, tolerability, and serum pharmacokinetics of CLG561 in subjects with AMD. A phase 2 study (NCT02515942) started on 2015 enrolling 114 participants, to evaluate the safety and efficacy of 12 (every 28 days) IVIs of CLG561 as a monotherapy and in combination with LFG316 as compared to sham in subjects with GA [26,41]. The study was completed, but the reporting date is on August 2020. 

#### 4.2.5. LFG316 

LFG316, or Tesidolumab, is a fully human IgG1 targeting complement factor C5 that inhibits the complement system activation [26,29,41]. A clinical phase 1 dose-escalation and safety study with single IVIs of 0.15–5mg LFG316 were performed in patients with GA or choroidal neovascularization due to AMD (NCT01255462). No adverse effects have been published and the drug was well tolerated. A phase 2 study (NCT02515942) started on 2015 enrolling 114 participants, to evaluate the safety and efficacy of 12 (every 28 days) IVIs of CLG561 in combination with LFG316 and CLG561 as a monotherapy and compared to sham in subjects with GA [29,30,42]. The studies were completed, but the reporting date is on August 2020. In a clinical trial phase 2 testing low doses of LFG316 (NCT01527500), 158 participants with GA were treated. The study was divided into 2 parts: part A evaluated the safety and efficacy of multiple 5 mg/50 µL doses of IVI LFG316 against sham every 28 days for 505 days and part B evaluated the safety and pharmacokinetics of a single IVI dose of 10 mg/100 µL of LFG316. At the completion of the trial, LFG-316 was found to have an acceptable safety profile, but was not effective in reducing GA lesion growth rate or improving visual acuity.

#### 4.2.6. Suppressors of Inflammation: Iluvien 

Iluvien is a sustained-release formulation of fluocinolone acetonide. It is a corticosteroid, a synthetic hydrocortisone derivative [31]. The fluorine substitution at position 9 in the steroid nucleus greatly enhances its activity. It is only approved for the treatment of diabetic macular edema (DME). Iluvien could slow the progression of GA. A total of 40 patients affected bilaterally by GA were recruited in a phase 2 study (NCT00695318). The study was completed, but the results are not yet available [31]. 

### 4.3. Anti-Oxidative Stress

#### Risuteganib (Luminate)

Risuteganib (Alg-1001) is an anti-integrin that downregulates oxidative stress and restores homeostasis. It localizes for several months in the RPE after the IVI giving luminate the potential to block the four pathways of oxidative stress. Alg-1001 targets three integrin receptors that are implicated in dry AMD in order to restore homeostasis in the retina. It aims to regulate oxidative stress before it has a chance to initiate multiple pathways of tissue damage. Currently, no phase 2 study results have been posted on a randomized controlled, double-masked, crossover clinical trial designed to evaluate the safety and exploratory efficacy of Risuteganib (1.0 mg), injected intravitreally, on eligible subjects who have been diagnosed with intermediate non-exudative AMD (Clinicaltrials.gov. NCT03626636) [32]. 

### 4.4. Other Treatment Modalities

#### Ocular Gene Therapy: AAVCAGsCD59

AAVCAGsCD59 (HMR59) is a potential IVI gene therapy for patients with GA, delivered via adeno-associated viral expression. The aim is to increase the expression of a soluble recombinant version of the naturally occurring CD59 (sCD59). sCD59 is designed to protect retinal cells by inhibiting the formation of the MAC, the terminal step of complement-mediated cell lysis. In gene therapy, the cells of the retina are permanently altered to produce sCD59 for whole duration of the patient’s life. With gene therapy a single injection of AAVCAGsCD59 is needed for the drug to be effective. The open-label and multi-center study of phase 1 will evaluate safety in 17 participants with additional 8 patients enrolled for 24 months. The 3 groups were divided according to 3 single intravitreal administrations at low, medium and high dosage in an office setting from March 2017 (NCT03144999). Currently, no study results have been posted on Clinicaltrials.gov [33].

### 4.5. Novel Compounds Derived from High-Throughput Drug Screens

#### RO7171009

RO7171009 or RG6147 is an investigational drug being evaluated for the potential treatment of GA secondary to AMD. The mechanism of action is yet undefined. The safety and tolerability of RO7171009 following single and multiple IVI administrations in patients with GA will be the object of an extra investigation through a phase 1, open-label, multicenter study. Currently, the study (NCT03295877) is composed of two levels of single-dose escalation (SAD), and multiple-dose (MD) stages in 28 patients. The examined results will be tolerability, safety, rate of adverse events, and serum concentration of RO7171009. The study ended on November 2018, and participants are no longer being examined or treated. Currently, no study results have been posted on Clinicaltrials.gov [34].

## 5. Discussion

There are still no treatments that slow progression of dry AMD. Ongoing clinical trials continue to pursue new drugs for the purpose of preventing and/or treating dry AMD, although some intitially promising complement pathway inhibitors (such as lampalizumab) have failed to achieve expected outcomes in clinical trials, a number of compelling prospects remain [13,14,15,16,17,18,19,20,21,22,23,24,25,26,27,28,29,30,31,35,36,37,38,39,40,41,42]. Currently, APL-2, which binds the C3 protein blocking all three pathways of complement activation, may be the most promising molecule [26,27,28,40,41]. The clinical trials Oaks and Derby, enrolling 600 patients each and ongoing at 58 different locations, will be completed in 2022 [26,27,28,40]. Similarly, we are awaiting the results on brimonidine, a neuroprotective agent, which will be published in 2019 [13,14].

No statistically significant improvement in visual acuity was observed across treatment groups using NT-501 ECT implantation to deliver CNTF via genetically engineered human cells [15,16,17,18,38,39]. It is possible to speculate that these molecules might be more effective if injected in the initial phases of the macular degeneration, especially because they act as neuroprotectors or immunomodulating/anti-inflammatory agents. We know that the mechanism of action of the neuroprotectors is to prevent degeneration and apoptosis of ganglion cells whilst the mechanism of action for the immunomodulating agents is to block the complement cascade model. The pharmacological association, alternating the IVIs during the administration of the therapeutic protocol, could be useful for the different mechanism of action of the two classes of molecules.

The main issue the researchers are called to resolve is when to start the treatment of dry AMD patients. We should be sure not to start too early to avoid aggressive therapies, nor too late for the high risk of losing most of the healthy tissue. Moreover, during more advanced stages of the pathology, new genetic therapies could be able to block the biomolecular dysfunction inducing the death of retinal cells. In the late stages, after the development of GA and cell death in the macular area, only new therapies using staminal cells might be able to regenerate lost tissues.

We hypothesize that distinguishing different stages of dry AMD is critical for establishing guidelines to employ the most effective therapeutic strategy for each patient [43,44,45,46]. Retinal morphological changes might be used to identify patients early in the course of AMD development who might still be at a reversible phase of the disease and, therefore, amenable to intervention. Minimum sizes to define atrophic areas vary; the commonly used Wisconsin Grading System includes lesions ≥175 μm in diameter. The definitions of non-exudative AMD in the international statistical classification of diseases are in Table 2 [46,47,48,49,50,51]. A recent consensus of retina international experts strongly recommends multimodal imaging to measure and classify the different forms of neovascular and non-neovascular-AMD [52]. So, color fundus photography, confocal fundus autofluorescence, confocal near-infrared reflectance, and high-resolution optical coherence tomography volume scans should be acquired at regular intervals throughout the clinical studies [52]. The use of the different therapies depending on the stage might avoid a consequent severe visual loss in old age.

In light of these considerations, we would like to recall our research in which subjects affected by GA have been treated with eye drops based on nerve GF (NGF) [53], or with autologous stem cell transplantation [54,55,56]. It has been ascertained, through the Regen Lab SA of tissue engineering in Switzerland (Swiss Biotech Association, available online: https://www.swissbiotech.org/), that the autograft produces numerous neurotrophic and angiotrophic GFs determining a beneficial impact in terms of visual acuity and retinal sensitivity, evaluated with microperimetry and ocular electrophysiology [55,56,57,58,59].

## 6. Conclusions

The future of dry AMD is one in which the most appropriate and personalized treatment will be employed for each patient. To this end, we expect that future studies resulting in successful clinical trials will entail collaborations between researchers studying disparate approaches to dry AMD.

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
