# Peer review of "Therapeutic Approaches with Intravitreal Injections in Geographic Atrophy Secondary to Age-Related Macular Degeneration: Current Drugs and Potential Molecules"

_ijms, 2019, doi:10.3390/ijms20071693_

Reviewer 1 Report

comment 1.

There is a side effect of Brimonidine Cartrate, but it does not appear in the journal, so it is necessary to add it.

Ref 1. Side-Effect Profile of Brimonidine Tartrate in Children.

Ophthalmology 2005;112: 21432148 © 2005 by the American Academy of Ophthalmology.

comment 2.

As  your report the POT-4 showed no safety concerns. However according to  the reference, phase 1 study of POT-4 in AMD eyes with CNV was completed successfully without any safety concerns at doses up to 1.05 mg  (NCT00473928).

So, I wonder that what safe concentration of phase 2 study of POT-4

Ref 2. Pharmacotherapy of Age-Related Macular Degeneration,Mark S. Blumenkranz, Marco A. Zarbin, in Retina (Fifth Edition), 2013

comment 3.

It  would be better to add information about neuroprotective agents of  serotonin 1A agonist(Tandospirone). This is approved and marketed in  japan, as an antidepressant. It has demonstrated neuroprotection in  animal models, showing dose-dependent protection of photoreceptors and  RPE cells from severe photo oxidative stress.

comment 4.

It would be better to add information about Anti-inflammation drugs.

According  to the reference Eculizumab, FCFD4514S and Glatiramer Acount is also  suppressed inflammation. These drug are clinical phase I~

This reference for comment 3 and 4.

Ref 3,4 Overview of Clinical Trials for Dry Agerelated Macular Degeneration, 2017 Journal of Medical Sciences | Published by Wolters Kluwer – Medknow, J Med Sci 2017;37(4):121129

comment 5.

It would be better to add information about clinical phase of Table 1. 

comment 6.

The target contents of the Iluvien drug are not visible of Table 1.

Author Response

Review Report Form     (Reviewer 1). Author's Reply to the Review Report

(x) English language and style are fine/minor spell check requie.

We have performed the English language check.

Comments and Suggestions for Authors

comment 1.

There is a side effect of Brimonidine Cartrate, but it does not appear in the journal, so it is necessary to add it.

Ref 1. Side-Effect Profile of Brimonidine Tartrate in Children.

Ophthalmology 2005;112: 2143–2148 © 2005 by the American Academy of Ophthalmology.

The side effect and the references were added (see page 7 and Ref. 14-15).

comment 2.

As  your report the POT-4 showed no safety concerns. However according to  the reference, phase 1 study of POT-4 in AMD eyes with CNV was completed successfully without any safety concerns at doses up to 1.05 mg  (NCT00473928).

So, I wonder that what safe concentration of phase 2 study of POT-4

Ref 2. Pharmacotherapy of Age-Related Macular Degeneration,Mark S. Blumenkranz, Marco A. Zarbin, in Retina (Fifth Edition), 2013.

The sentence and the references were added: “A phase I clinical trial (NCT00473928) in wet AMD patients was completed in 2010 without safety concerns at doses up to 1.05 mg [31-33]”

comment 3.

It  would be better to add information about neuroprotective agents of  serotonin 1A agonist (Tandospirone). This is approved and marketed in  Japan, as an antidepressant. It has demonstrated neuroprotection in  animal models, showing dose-dependent protection of photoreceptors and  RPE cells from severe photo oxidative stress. 

We apologize to you, but Tandospirone is an Oftalmic Solution and not an intravitreal injection; for the moment it cannot be included in the manuscript, otherwise we would have had to insert many other molecules. Moreover, Clinical trial (NCT00890097) ended July 3, 2014, but the Treatment was ineffective.

comment 4.

It would be better to add information about Anti-inflammation drugs.

This reference for comment 3 and 4.

Ref 3,4 Overview of Clinical Trials for Dry Age‑related Macular Degeneration, 2017 Journal of Medical Sciences | Published by Wolters Kluwer – Medknow, J Med Sci 2017;37(4):121‑129.

We added the Reference (see n. 59).

comment 5.

It would be better to add information about clinical phase of Table 1.    

We have already included the clinical trial phase on table 1 since the beginning of the manuscript (see Table 1, Column 4, Studies, and notes. Ph 1,2,3=Phase 1,2,3).

comment 6.

The target contents of the Iluvien drug are not visible of Table 1.

We have eliminated the defect in Table 1.

Reviewer 2 Report

The review paper “Therapeutic approaches with intravitreal injections in geographic atrophy secondary to age-related macular degeneration: current drugs and potential molecules” by Nebbioso et al., is a short article specifically focused on intravitreal therapies progressing through the clinical pipeline. Despite the article being short and mostly easy to read, I found it inconsistent and meandering. For instance, the authors included a major section titled  “Current Developments in Intravitreal Therapy”. But rather than describing current developments in this section, the authors very briefly describe the complement protein pathway (only one out of several therapeutic targets). Consequently I felt the text of the section did not logically follow the section heading, and the section conclusion—that the blood-retinal barrier complicates drug delivery (which is why the authors focus on IVT therapies)—did not logically follow the text.  Although other sections have a more logical flow, there are many instances of repetitious text,  typos, and superfluous statements. The discussion likewise needs to be restructured.

Another problem is that the authors fail to clearly outline the coherent link between the drug therapy, the target, and the target’s relationship to geographic atrophy.  For example, in section 4.1.1., the authors explain that Brimonidine is neuroprotective in glaucoma, that it modulates Aβ toxicity, but they entirely fail to clarify why Brimonidine might be effective for preventing the RPE degeneration during geographic atrophy.  This needs to be tightened up and clarified for every drug! 

The authors are also inconsistent about which therapeutic approaches they decide to include in their review. Entire paragraphs are devoted to Lampalizumab and to LFG316, both of which were found to be ineffective treatments for geographic atrophy.  However, other drugs were not covered. I did a brief search in clinicaltrials.gov for geographic atrophy, and was surprised to find ongoing trials for geographic atrophy using IVT that are not included in this review, such as: AAVCAGsCD59 (HMR59 NCT03144999), Luminate (Alg-1001, risuteganib NCT03626636), IONIS-FB-Lrx NCT03815825, and RO7171009, NCT03295877).

Additionally, the authors use the discussion to highlight their own research of therapies delivered via eyedrops (NGF) or scleral pocket autograft (stem cells), neither of which is consistent with the review topic of IVT therapies in the clinical pipeline.  This entire paragraph is both out of place and inappropriate for the discussion section of a review.

Finally, Scheme 1 is fairly impenetrable, and the figure text doesn’t give any additional insight. A quick google search for complement pathway images brings up several similar figures that I found to be much more “user friendly”. Moreover, the three figures are blurry and need to be saved in a different format.

Author Response

Review Report Form     (Reviewer 2). Author's Reply to the Review Report

Moderate English changes requie.

We have performed the English language check.

Comments and Suggestions for Authors

The review paper “Therapeutic approaches with intravitreal injections in geographic atrophy secondary to age-related macular degeneration: current drugs and potential molecules” by Nebbioso et al., is a short article specifically focused on intravitreal therapies progressing through the clinical pipeline. Despite the article being short and mostly easy to read, I found it inconsistent and meandering. For instance, the authors included a major section titled  “Current Developments in Intravitreal Therapy”. But rather than describing current developments in this section, the authors very briefly describe the complement protein pathway (only one out of several therapeutic targets). Consequently I felt the text of the section did not logically follow the section heading, and the section conclusion—that the blood-retinal barrier complicates drug delivery (which is why the authors focus on IVT therapies)—did not logically follow the text.  Although other sections have a more logical flow, there are many instances of repetitious text,  typos, and superfluous statements. The discussion likewise needs to be restructured. Recently, evidence has implicated an overactive inflammatory cascade called the complement system as playing a pivotal role in the development of dry AMD. Another problem is that the authors fail to clearly outline the coherent link between the drug therapy, the target, and the target’s relationship to geographic atrophy.  For example, in section 4.1.1., the authors explain that Brimonidine is neuroprotective in glaucoma, that it modulates Aβ toxicity, but they entirely fail to clarify why Brimonidine might be effective for preventing the RPE degeneration during geographic atrophy.  This needs to be tightened up and clarified for every drug!  

The manuscript was completely revised and the section title “Current Developments in Intravitreal Therapy” changed to: “Main pathogenetic pathways in dry AMD: Complement dysfunction and inflammation”.

The reviewer states: "Recently, evidence has implicated an overactive inflammatory cascade called the complement system as playing a pivotal role in the development of dry AMD." In fact, we have taken this concept from recent literature and inserted it into the manuscript as the main element of the disease.

In this context, we ophthalmologists have reported in the paper the molecules currently studied in clinical trials by researchers. Nowadays, we  don't yet know the specific biochemical action of several of these molecules on the ocular tissues.

The field of studies is open and it needs extensive research before being able to find useful drugs for patients with dry AMD. Today we don't have effective therapy and our patients lose their vision, while we are impotent to evolution in geographic atrophy.

The authors are also inconsistent about which therapeutic approaches they decide to include in their review. Entire paragraphs are devoted to Lampalizumab and to LFG316, both of which were found to be ineffective treatments for geographic atrophy.  However, other drugs were not covered. I did a brief search in clinicaltrials.gov for geographic atrophy, and was surprised to find ongoing trials for geographic atrophy using IVT that are not included in this review, such as: AAVCAGsCD59 (HMR59 NCT03144999), Luminate (Alg-1001, risuteganib NCT03626636), IONIS-FB-Lrx NCT03815825, and RO7171009, NCT03295877).

The most known and used molecules in AMD dry are those of which we have more news and we have inserted them for what is present in the international literature (see also previous answer).

We thank the reviewer and added only the drugs used for intravitreal injection: NCT03144999                 NCT03626636                NCT03295877

Additionally, the authors use the discussion to highlight their own research of therapies delivered via eyedrops (NGF) or scleral pocket autograft (stem cells), neither of which is consistent with the review topic of IVT therapies in the clinical pipeline.  This entire paragraph is both out of place and inappropriate for the discussion section of a review.

In the Discussion Section we would like to expose our ideas and give strength to what we think is right. Readers will be able to judge current research useful for dry AMD and GA patients according to the severity of the case. We have significantly shortened the paragraph.

Finally, Scheme 1 is fairly impenetrable, and the figure text doesn’t give any additional insight. A quick google search for complement pathway images brings up several similar figures that I found to be much more “user friendly”. Moreover, the three figures are blurry and need to be saved in a different format. 

The scheme is original and we cannot use those on the web. The figures are temporary. They will be sent in an appropriate format when the Editorial Office requests them.

Round  2

Reviewer 2 Report

The authors have made a concerted effort to improve the manuscript and provide additional details that were lacking in the previous version.  The current version still needs minor editing, mostly for language and clarity. Please see my edits here, and if the authors are willing make these improvements, I will accept it for publication.  Please note I did not correct any of the numerous punctuation problems, which I expect to be addressed by the publishers own editors.

PAGE 1:

Line 19: change  “…and agents called neuroprotective.” to …and neuroprotective agents.

Line 20: change “…block the complement cascade for immunomodulating agents” to “block complement cascade upregulation of immunomodulating agents”

Lines 22-26: This passage is overwritten and meaningless.  Please delete entire passage, starting with: “To the best of knowledge, and after extensive studies on the matter....” Please replace with a concluding remark such as “Our analysis indicates that finding treatments for dry AMD will require continued collaboration among researchers to identify additional molecular targets and to fully interrogate the utility of pluripotent stem cells for personalized therapy.”

Line 35: change “and, probably, there will be…” to “with probable increases to 196 million …”

Line 36: change “…in early stage of…” to “…at the early stage of…”

PAGE 2:

Line 5: change “…treatment are effective…” to “treatments are effective…”

Line 8: Delete the entire sentence “Nowadays, new therapies are under investigation, but unfortunately,  no valid drugs are currently available for dry AMD.” (You basically stated this already in lines 4-5.) . Start with: “Currently, the only preventive option for dry AMD is the Age-Related Eye Disease ….”

Line 11: change “…the resultant of…” to “the result of…”

Lines 17-18: Delete “…in the disease.”

PAGE 3:

Line 45-46: Please delete “, a current hot topic for international literature and the scientific world.” This is a meaningless statement.

PAGE 4:

Line 19: change “…in order to avoid, as far as possible, the final negative visual prognosis in GA” to “ …and preserve vision”

Lines 19-20: Change “With these premises, the present review will be focused on.…” to “The present review will focus on…”

Lines 21-22: Change “The summary of clinical trials for the….” To “A summary of relevant clinical trials is presented in Table 1.”

PAGE 7:

Lines 9-10: Change “..brimonidine preserves from degeneration photoreceptors, bipolar cells,…” to “…brimonidine protects photoreceptors, bipolar cells, and retinal ganglion cells from a variety of retinal insults, including…”

Lines 10-13: Change “…the neuroprotective action probably takes place in different ways including the regulation of the neurotrophic ……and through models of intrinsic survival cell signaling” to “…neuroprotection is achieved via different mechanisms such as the release of brain derived neurotrophic factor (BDNF) from retinal ganglion cells and by upregulation of cell survival signaling pathways.”

Line 13: Change “In short, brimonidine has been….” To “Importantly, brimonidine has been….”

Lines 27-28: Change “…and therefore promote cell…” to “…thereby promoting cell….”

Line 29: Delete “…after a 6-months follow up,”

Line 30: Change “…determined a significant reduction in…” to “…produced a significant reduction in…”

Line 31: change “200 ug dose determined...” to  “the 200 ug dose produced…”

Line 32: change “…while 400 ug by 27%.” To “…while the 400 ug dose produced a 27% reduction.”

Line 32: what do you mean by “inserts”? do you mean “injections?”

PAGE 8:

Line 3: change “This could be…” to “This could….”

Line 12: Delete “and released in situ”

Line 13: Change “…the CNTF” to “CNTF”

Line 30: Change “…associated to visual…” to “…associated with visual…”

Line50: Change “…to receive sham every…” to “to receive sham injections every…”

PAGE 9:

Line 11: add “A” to A phase 1 trial for dry AMD…”

Line 20: add “patients” to “estimated one hundred-twenty patients

Line 21: change “to sham in subjects…” to “to sham injection in subjects…”

Line 30: change “and prevent MAC…” to “and preventing MAC”

Line 43: “12 months” should be “12 month”

PAGE 10:

Line 24: change “Study was divided…” to “The study was divided…”

Line 40: change “…effect to block the four…” to “to block the four…”

PAGE 11:

Line 1: change “…with intermediate of non-exudative…” to “…with intermediate non-exudative...”

Line 3: 4.4 Biological Treatment:      All of the treatments covered in the review are biological treatments.  Please change the heading to “Other Treatment Modalities” or something similar.

Line 5: Please change first sentence to “AAVCAGsCD59 is a potential IVT gene therapy for patients with GA, delivered via adeno-associated viral expression.”

Line 7: Delete “The” in “The sCD59 is designed…”

Line 15: Please change “New Molecular Entity” to “Novel Compounds Derived from High Throughput Drug Screens” I think this is more informative as a subcategory.

Line 27: Delete “Nowadays” it is a colloquial term, not suitable here.

Lines 27-32: Change “Some clinical trials are ongoing, with the goal of…..the pathology. Several new drugs have shown some promising results…..but it was disproved by Chroma and Spectri during phase 3 clinical trials”      to   “Ongoing clinical trials continue to pursue new drugs for the purpose of preventing and/or treating dry AMD, Although some intitially promising complement pathway inhibitors (such as lampalizumab) have failed to achieve expected outcomes in clinical trials, a number of compelling prospects remain. Currently, APL-2….”

Lines 38-39: change “…after implant of CNTF with NT-501 ECT,….therapeutic protein in situ.” to “…using NT-501 ECT implantation to deliver CNTF via genetically engineered human cells.”

PAGE 12:

Lines 2-3: “An association and complementary pharmacological use may be suggested given the different action mechanism of 3 the two classes of molecules.” Please rephrase. I do not understand.

Lines 8-9: Change “…could be able to block the processes that create a biomolecular malfunction inducing…” to  “…could be able to block the biomolecular dysfunction inducing…”

Lines 12-13: Change “On the other hand, we hypothesize…of the guidelines in the differentially affected patients” to “We hypothesize that distinguishing different stages of dry AMD is critical for establishing guidelines to employ the most effective therapeutic strategy for each patient.”

Line 14: Change “The retinal morphological changes…” to “Retinal morphological changes…”

Lines 19-20: Change “…has highlighted how multimodal…is strongly recommended to….” to “…strongly recommends multimodal imaging to measure and classify …-AMD.”

PAGE 13: Conclusions

Lines 1-6: The Conclusion, like the last two sentences of the abstract is weak, poorly worded and repetitive.  I recommend something brief and to the point! Such as:  “The future of dry AMD is one in which the most appropriate and personalized treatment will be employed for each patient.  To this end, we expect that future studies resulting in successful clinical trials will entail collaborations between researchers studying disparate approaches to dry AMD.”

Author Response

March 31, 2019

Dear Guest Editor

Dr. Janusz Błasiak

We are sending you the revised manuscript, ID: ijms-453126 (REV 2),  titled: Therapeutic approaches with intravitreal injections in geographic atrophy secondary to age-related macular degeneration: current drugs and potential molecules, by Marcella Nebbioso, Alessandro Lambiase, Alberto Cerini, Paolo Giuseppe Limoli, Maurizio La Cava, and Antonio Greco Nicola Pescosolido*, Francesco Parisi**, Giuseppe Buompristhat has been reviewed according to the comments. Changes made in response to comments of the reviewer are included at the bottom of this letter.

Many thanks

                                                                                            Marcella Nebbioso & Co-Authors     

Address of Corresponding Authors

Marcella Nebbioso, Department of Sense Organs, Ocular Electrophysiology Center, Policlinico Umberto I, Sapienza University of Rome, Rome, piazzale Aldo Moro 5, 00185, Italy. e-mail: marcella.nebbioso@uniroma1.it Tel.: +39-06-49975422;  Fax: +39-06-49975425.

COMMENTS TO AUTHORS AND ANSWERS

Editorial comments:

After plagiarism check for your manuscript, we found some sentences share high similarity with published papers. File attached is the plagiarism report. Please rewrite the highlighted sentences with track-change (The similarity index should be less than 5%). Thanks for your cooperation. 

We have modified the sentences.

Review Report Form. Author's Reply to the Review Report

The authors have made a concerted effort to improve the manuscript and provide additional details that were lacking in the previous version.  The current version still needs minor editing, mostly for language and clarity. Please see my edits here, and if the authors are willing make these improvements, I will accept it for publication.  Please note I did not correct any of the numerous punctuation problems, which I expect to be addressed by the publishers own editors.

We thank the Referee for the recommendations. Thus, we have changed and improved the manuscript as required. All the improvements have been made and thanks again for the help.

PAGE 1:  Line 19: change  “…and agents called neuroprotective.” to …and neuroprotective agents

Line 20: change “…block the complement cascade for immunomodulating agents” to “block complement cascade upregulation of immunomodulating agents” 

Lines 22-26: This passage is overwritten and meaningless.  Please delete entire passage, starting with: “To the best of knowledge, and after extensive studies on the matter....” Please replace with a concluding remark such as “Our analysis indicates that finding treatments for dry AMD will require continued collaboration among researchers to identify additional molecular targets and to fully interrogate the utility of pluripotent stem cells for personalized therapy.” 

Line 35: change “and, probably, there will be…” to “with probable increases to 196 million …” 

Line 36: change “…in early stage of…” to “…at the early stage of…” 

PAGE 2:  Line 5: change “…treatment are effective…” to “treatments are effective…” 

Line 8: Delete the entire sentence “Nowadays, new therapies are under investigation, but unfortunately,  no valid drugs are currently available for dry AMD.” (You basically stated this already in lines 4-5.) . Start with: “Currently, the only preventive option for dry AMD is the Age-Related Eye Disease ….” 

Line 11: change “…the resultant of…” to “the result of…” 

Lines 17-18: Delete “…in the disease.” 

PAGE 3:   Line 15-16: Please delete “, a current hot topic for international literature and the scientific world.”This is a meaningless statement.

PAGE 4:  Line 19: change “…in order to avoid, as far as possible, the final negative visual prognosis in GA”to “ …and preserve vision” 

Lines 19-20: Change “With these premises, the present review will be focused on.…” to “The present review will focus on…” 

Lines 21-22: Change “The summary of clinical trials for the….” To “A summary of relevant clinical trials is presented in Table 1.” 

PAGE 7:  Lines 9-10: Change “..brimonidine preserves from degeneration photoreceptors, bipolar cells,…”to “…brimonidine protects photoreceptors, bipolar cells, and retinal ganglion cells from a variety of retinal insults, including…” 

Lines 10-13: Change “…the neuroprotective action probably takes place in different ways including the regulation of the neurotrophic ……and through models of intrinsic survival cell signaling” to “…neuroprotection is achieved via different mechanisms such as the release of brain derived neurotrophic factor (BDNF) from retinal ganglion cells and by upregulation of cell survival signaling pathways.” 

Line 13: Change “In short, brimonidine has been….” To “Importantly, brimonidine has been….” 

Lines 27-28: Change “…and therefore promote cell…” to “…thereby promoting cell….” 

Line 29: Delete “…after a 6-months follow up,” 

Line 30: Change “…determined a significant reduction in…” to “…produced a significant reduction in…” 

Line 31: change “200 ug dose determined...” to  “the 200 ug dose produced…” 

Line 32: change “…while 400 ug by 27%.” To “…while the 400 ug dose produced a 27% reduction.” 

Line 32: what do you mean by “inserts”? do you mean “injections?” 

PAGE 8:   Line 3: change “This could be…” to “This could….” 

Line 12: Delete “and released in situ” 

Line 13: Change “…the CNTF” to “CNTF” 

Line 30: Change “…associated to visual…” to “…associated with visual…” 

Line 50: Change “…to receive sham every…” to “to receive sham injections every…” 

PAGE 9:   Line 11: add “A” to A phase 1 trial for dry AMD…” 

Line 20: add “patients” to “estimated one hundred-twenty patients 

Line 21: change “to sham in subjects…” to “to sham injection in subjects…” 

Line 30: change “and prevent MAC…” to “and preventing MAC” 

Line 43: “12 months” should be “12 month” 

PAGE 10:   Line 24: change “Study was divided…” to “The study was divided…” 

Line 40: change “…effect to block the four…” to “to block the four…” 

PAGE 11:   Line 1: change “…with intermediate of non-exudative…” to “…with intermediate non-exudative...” 

Line 3: 4.4 Biological Treatment:      All of the treatments covered in the review are biological treatments.  Please change the heading to “Other Treatment Modalities” or something similar. 

Line 5: Please change first sentence to “AAVCAGsCD59 is a potential IVT gene therapy for patients with GA, delivered via adeno-associated viral expression.” 

Line 7: Delete “The” in “The sCD59 is designed…” 

Line 15: Please change “New Molecular Entity” to “Novel Compounds Derived from High Throughput Drug Screens” I think this is more informative as a subcategory. 

Line 27: Delete “Nowadays” it is a colloquial term, not suitable here. 

Lines 27-32: Change “Some clinical trials are ongoing, with the goal of…..the pathology. Several new drugs have shown some promising results…..but it was disproved by Chroma and Spectri during phase 3 clinical trials”      to   “Ongoing clinical trials continue to pursue new drugs for the purpose of preventing and/or treating dry AMD, Although some intitially promising complement pathway inhibitors (such as lampalizumab) have failed to achieve expected outcomes in clinical trials, a number of compelling prospects remain. Currently, APL-2….” 

Lines 38-39: change “…after implant of CNTF with NT-501 ECT,….therapeutic protein in situ.” to “…using NT-501 ECT implantation to deliver CNTF via genetically engineered human cells.” 

PAGE 12:  Lines 2-3: “An association and complementary pharmacological use may be suggested given the different action mechanism of 3 the two classes of molecules.” Please rephrase. I do not understand.  

Lines 8-9: Change “…could be able to block the processes that create a biomolecular malfunction inducing…” to  “…could be able to block the biomolecular dysfunction inducing…” 

Lines 12-13: Change “On the other hand, we hypothesize…of the guidelines in the differentially affected patients” to “We hypothesize that distinguishing different stages of dry AMD is critical for establishing guidelines to employ the most effective therapeutic strategy for each patient.” 

Line 14: Change “The retinal morphological changes…” to “Retinal morphological changes…” 

Lines 19-20: Change “…has highlighted how multimodal…is strongly recommended to….” to “…strongly recommends multimodal imaging to measure and classify …-AMD.” 

PAGE 13: Conclusions    Lines 1-6: The Conclusion, like the last two sentences of the abstract is weak, poorly worded and repetitive.  I recommend something brief and to the point! Such as:  “The future of dry AMD is one in which the most appropriate and personalized treatment will be employed for each patient.  To this end, we expect that future studies resulting in successful clinical trials will entail collaborations between researchers studying disparate approaches to dry AMD.”